# TAZ links exercise to mitochondrial biogenesis via mitochondrial transcription factor A

Jun-Ha Hwang[1], Kyung Min Kim [1], Ho Taek Oh[1], Gi Don Yoo[1], Mi Gyeong Jeong[2], Hyun Lee[1], Joori Park [1], Kwon Jeong[1], Yoon Ki Kim [1], Young-Gyu Ko [1], Eun Sook Hwang [2✉] & Jeong-Ho Hong [1✉]

Mitochondria are energy-generating organelles and mitochondrial biogenesis is stimulated to meet energy requirements in response to extracellular stimuli, including exercise. However, the mechanisms underlying mitochondrial biogenesis remain unknown. Here, we demonstrate that transcriptional coactivator with PDZ-binding motif (TAZ) stimulates mitochondrial biogenesis in skeletal muscle. In muscle-specific TAZ-knockout (mKO) mice, mitochondrial biogenesis, respiratory metabolism, and exercise ability were decreased compared to wild-type mice. Mechanistically, TAZ stimulates the translation of mitochondrial transcription factor A via Ras homolog enriched in brain (Rheb)/Rheb like 1 (Rhebl1)-mTOR axis. TAZ stimulates *Rhebl1* expression via TEA domain family transcription factor. Rhebl1 introduction by adeno-associated virus or mTOR activation recovered mitochondrial biogenesis in mKO muscle. Physiologically, mKO mice did not stimulate exercise-induced mitochondrial biogenesis. Collectively, our results suggested that TAZ is a novel stimulator for mitochondrial biogenesis and exercise-induced muscle adaptation.

[1] Department of Life Sciences, School of Life Sciences and Biotechnology, Korea University, Seoul 02841, Korea. [2] College of Pharmacy, Ewha Womans University, Seoul 03760, Korea. ✉email: eshwang@ewha.ac.kr; jh_hong@korea.ac.kr

Mitochondrial biogenesis is a process by which cells increase their mitochondrial mass and copy number; it is regulated in response to extracellular stimuli, including nutrients, hormones, and exercise[1–4]. Diverse signals for the stimulation of mitochondrial biogenesis converge on transcriptional regulators, including peroxisome proliferator-activated receptor gamma coactivator 1-alpha (PGC1α)[5,6]. PGC1α interacts with several transcription factors, including NRF1 and NRF2, to stimulate the genes responsible for mitochondrial biogenesis[7,8]. NRF1 and NRF2 stimulate the expression of mitochondrial transcription factor A (TFAM) and transcription factor B proteins, which are the major regulators of mitochondrial DNA transcription and replication[8,9].

Metabolic disorders such as obesity and type 2 diabetes, resulting from sedentary lifestyles, are common in modern society. Exercise over both the short and long term is prescribed for the treatment of metabolic disorders[10,11], and regular exercise with dietary restriction has demonstrated improvements in therapeutic outcomes for metabolic disorders compared to pharmacological treatment[11]. Particularly, exercise promotes mitochondrial biogenesis, leading to increased metabolic rate, energy expenditure, and fat utilization[12]. This increases whole-body metabolism and helps prevent metabolic disorders. Repeated bouts of muscle contraction during exercise induce muscle adaptation, including changes in intracellular signalling, mitochondrial mass and function, and metabolic regulation[13–15]. The changes reflect the exercise-mediated activation of specific signalling pathways that regulate the transcription and translation of exercise-responsive genes. Although the benefits of exercise to health are well-known, the underlying mechanisms, including exercise-induced mitochondrial biogenesis, are not well-understood.

Hippo signalling is a key regulator of cell growth and differentiation, which are bioenergy-intensive processes. Hippo signalling cascades regulate the transcriptional coregulators, namely, transcriptional coactivator with PDZ-binding motif (TAZ) and Yes-associated protein (YAP) that interact with several transcription factors, including TEA domain (TEAD) family members, and regulate their target genes in response to Wnt, G protein-coupled receptor signalling, and mechanotransduction[16–19]. Moreover, TAZ/YAP activity is regulated by exercise-associated signals[20]. However, whether these transcriptional co-regulators play roles in mitochondrial biogenesis is not known.

Here, we demonstrate that TAZ stimulates mitochondrial biogenesis via translational induction of Tfam and is responsible for exercise-mediated muscle adaptation.

## Results

**TAZ knockout in muscle decreases mitochondrial mass, respiration, and exercise ability.** Skeletal muscle functions as a biological motor for organismal movements, utilizes ATP, and exhibits a large number of mitochondria to fulfill its high energy demands. To analyse the role of TAZ in muscle, muscle-specific TAZ-knockout (KO) mice models (*MCK-cre*;*Taz^f/f*, mKO) were developed[21]. We observed that the muscle mass of the mKO mice was similar to that of wild-type (WT) mice[21]. We further analysed WT and mKO muscle tissues via transmission electron microscopy. As shown in Fig. 1a, the total number of mitochondria and their corresponding total area were substantially decreased in mKO mice. Moreover, the protein levels of Cytochrome C, Cox4, succinate dehydrogenase complex flavoprotein subunit A (Sdha), and voltage-dependent anion-selective channel (Vdac) were also decreased in mKO mice (Fig. 1b). Subsequently, using WT and mKO mice, we investigated respiratory metabolism and activity under normal physiological conditions, as well as exercise ability under forced movement. To analyse respiratory

metabolism and activity under normal conditions, WT and mKO mice were acclimated to metabolic cages for 3 days, following which $O_2$ consumption and $CO_2$ production were analysed for 1 day. In addition, mouse activity was also measured for 1 day using an infrared-based movement sensor and weight transducer. Interestingly, compared to WT mice, mKO mice demonstrated significantly reduced $O_2$ consumption, $CO_2$ production, and energy expenditure at night (Fig. 1c). The rearing and movement counts were also decreased at night (Fig. 1d). Food and drink intake were similar between the WT and mKO mice. However, food intake was marginally lower in mKO mice during daytime (Fig. 1e). With respect to forced movements, mKO mice exhibited decreased exercise ability during the rotarod test (Fig. 1f) and shorter running distances during the treadmill endurance test (Fig. 1g). These results indicate that TAZ depletion in skeletal muscle impairs mitochondrial biogenesis, respiratory metabolism, and exercise ability in mice.

To study the role of TAZ in mitochondrial biogenesis in vitro, WT and TAZ KO mouse embryonic fibroblasts (MEF) were labelled with MitoTracker, a mitochondrion-selective probe that accumulates in active mitochondria. As shown in Supplementary Fig. 1a, the TAZ KO MEF demonstrated decreased staining intensity than that by WT MEF, indicating lowered mitochondrial activity. A reduction in the mitochondrial DNA copy number was also observed (Supplementary Fig. 1b). We also investigated the mitochondrial function. Mitochondria generate ATP via electrochemical forces[3]. Production of ATP (Supplementary Fig. 1c) and reactive oxygen species (Supplementary Fig. 1d) was decreased in TAZ KO MEF. Similarly, mitochondrial activity (Supplementary Fig. 1e), DNA copy number (Supplementary Fig. 1f), and ATP production (Supplementary Fig. 1g) were also decreased in C2C12 myoblasts with TAZ knockdown (KD). We also analysed the cellular oxygen consumption rate (OCR) of WT and TAZ KO MEF. Compared to WT cells, TAZ KO MEF exhibited decreased basal respiration under normal conditions along with a reduced maximal respiratory capacity after treatment with the mitochondrial uncoupler, carbonyl cyanide-4-(trifluoromethoxy)phenylhydrazone (FCCP) (Supplementary Fig. 1h). A similar result was observed in C2C12 myotubes (Supplementary Fig. 1i). These results demonstrate that TAZ plays an important role in mitochondrial biogenesis in vitro.

Next, we investigated whether TAZ depletion in adult muscle decreases mitochondrial biogenesis using TAZ shRNA containing Adeno-Associated Virus serotype 6 (AAV6). It has been shown that AAV6 mediates efficient transduction into muscle tissue[22,23]. The virus was directly injected into adult muscle of mice to deplete endogenous TAZ. As shown in Supplementary Fig. 2a, TAZ levels were decreased in TAZ shRNA virus-transduced muscle. The protein levels of Cox4, Sdha, Vdac, and Cytochrome C were also decreased in TAZ shRNA virus-transduced muscle (Supplementary Fig. 2a). Decreased expression of cytochrome c oxidase subunit 2 was further verified in TAZ shRNA virus-transduced muscle by immunofluorescent staining (Supplementary Fig. 2b). Mitochondrial DNA copy number was also decreased in TAZ shRNA virus-transduced muscle (Supplementary Fig. 2c). To verify the specificity of TAZ shRNA, we rescued WT TAZ or mutant TAZ (TAZ S51A), which is unable to bind with TEA domain family transcription factor (TEAD), in TAZ-depleted gastrocnemius muscle by AAV6-mediated transduction (Supplementary Fig. 2d). As shown in Supplementary Fig. 2e, f, restoration of mitochondrial protein level and DNA copy number were observed in WT TAZ rescued gastrocnemius muscle. However, the restoration was not significant in TAZ S51A rescued muscle (Supplementary Fig. 2e, f). These results suggested that impairment of mitochondrial biogenesis in mKO mice is not an event driven by developmental phenotype.

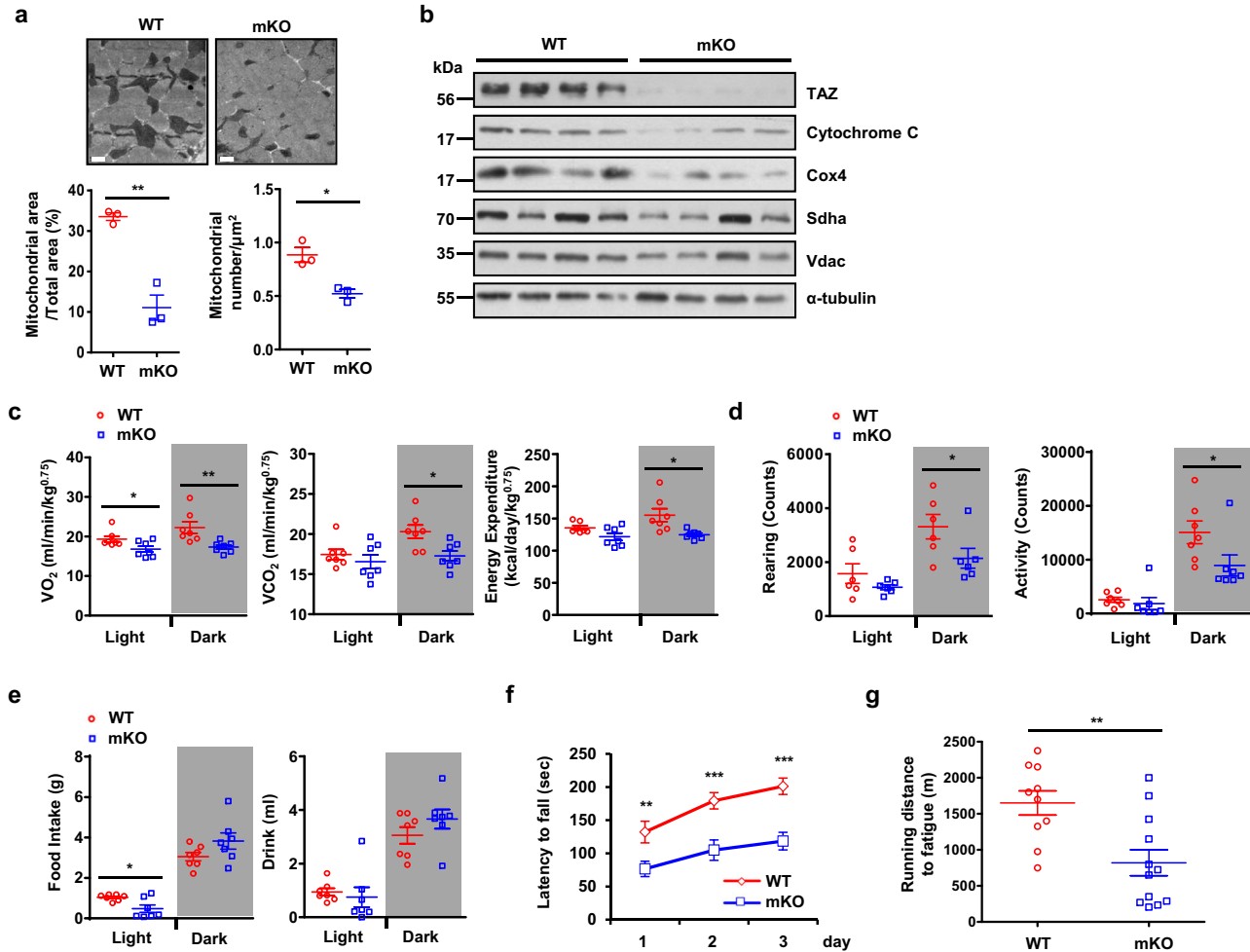

**Fig. 1 TAZ knockout in muscle decreases mitochondrial mass, respiration, and exercise ability. a** Images of the gastrocnemius muscle of wild-type (WT) or muscle-specific TAZ knockout (mKO) mice were acquired using a transmission electron microscope; the mitochondrial area or number of mitochondria per μm$^2$ was counted and calculated using Image J. Scale bar = 0.5 μm, $n = 3$ for each genotype (Mitochondrial area; *$p = 0.022$, mitochondrial number; *$p = 0.0104$). **b** Proteins of the gastrocnemius muscle of WT and mKO mice were analysed via immunoblotting for the detection of mitochondrial marker proteins. Alpha-tubulin was used as the loading control. Representative data was shown and the experiment was performed twice with similar results. **c** The respiratory and metabolic parameters of WT and mKO mice were assessed via indirect calorimetry tests. Data are shown as average values; $n = 7$ for each group (VO$_2$; *$p = 0.0353$ for light, **$p = 0.0089$ for dark, VCO$_2$; *$p = 0.0134$, Energy expenditure; *$p = 0.0137$). **d** The activity of mice under normal physiological conditions was analysed using a metabolic cage; $n = 7$ for each group (Rearing; *$p = 0.036$, Activity; *$p = 0.0267$). **e** Food and water consumption of mice described in (**a**) were monitored with a metabolic cage; $n = 7$ for each group (Food intake; *$p = 0.014$). **f** The exercise ability of WT and mKO mice was analysed using the rotarod test; $n = 8$ for WT mice and $n = 10$ for mKO mice (1 day; **$p = 0.00789$, 2 day; ***$p = 0.00158$. 3 day; ***$p = 0.00022$). **g** Endurance test of WT and mKO mice was conducted using a rodent treadmill; $n = 10$ for WT mice and $n = 12$ for mKO mice (***$p = 0.0017$). Eight to ten-week-old mice were used. Data are presented as mean ± SEM for (**a**, **c**, **d**–**g**). Statistical significance was analysed using two-sided $t$-test for (**a**, **c**, **e**–**g**) and one-sided $t$-test for (**d**). Source data are provided as a Source data file.

**TAZ induces Tfam expression at the translational level**. We next studied whether transcriptional regulatory factors involved in mitochondrial biogenesis are downregulated by TAZ depletion. As shown in Fig. 2a, transcript level of *Tfam* and *Nrf1* was marginally decreased in mKO mice, whereas Tfam protein level was significantly decreased in the skeletal muscle of these mice (Fig. 2b). However, no significant difference was observed in PGC1α, NRF1, or NRF2 levels. Similarly, in C2C12 myotubes, *Tfam* expression was decreased, whereas that of *Ppargc1a*, *Nrf1*, and *Nrf2* was not altered significantly (Fig. 2c). The Tfam protein level was also significantly decreased in TAZ KD myotubes (Fig. 2d), with comparable results being observed in TAZ KO MEF (Fig. 2e, f). Importantly, the Tfam level was restored in TAZ-rescued TAZ KO MEF without corresponding alterations in PGC1α and Nrf1 levels (Fig. 2f). The restoration of Tfam levels was not attributed to *Tfam* expression as TAZ-rescued KO MEF

did not demonstrate significant *Tfam* expression (Fig. 2e). These results indicate that TAZ induces Tfam level in the post-transcriptional state for mitochondrial biogenesis. Next, we studied whether TAZ regulates Tfam translation. Mammalian target of rapamycin complex 1 (mTORC1) regulates mitochondrial biogenesis by selectively promoting the translation of mitochondrial genes via inhibition of eukaryotic translation initiation factor 4E (eIF4E)-binding proteins (4E-BPs)[24]. Therefore, we assessed the activity of 4E-BP1 and ribosomal S6 kinase (S6K) that phosphorylate ribosomal S6 for translational stimulation[25]. Indeed, the phosphorylated forms of 4E-BP1 and S6K were significantly reduced in mKO mice (Fig. 2g), TAZ KD C2C12 myotubes (Fig. 2h), and TAZ KO MEF (Fig. 2i), showing that translational processing is altered with TAZ depletion. In addition, the introduction of TAZ in TAZ KO MEF restored 4E-BP1- and S6K phosphorylation (Fig. 2i). These results suggest that TAZ

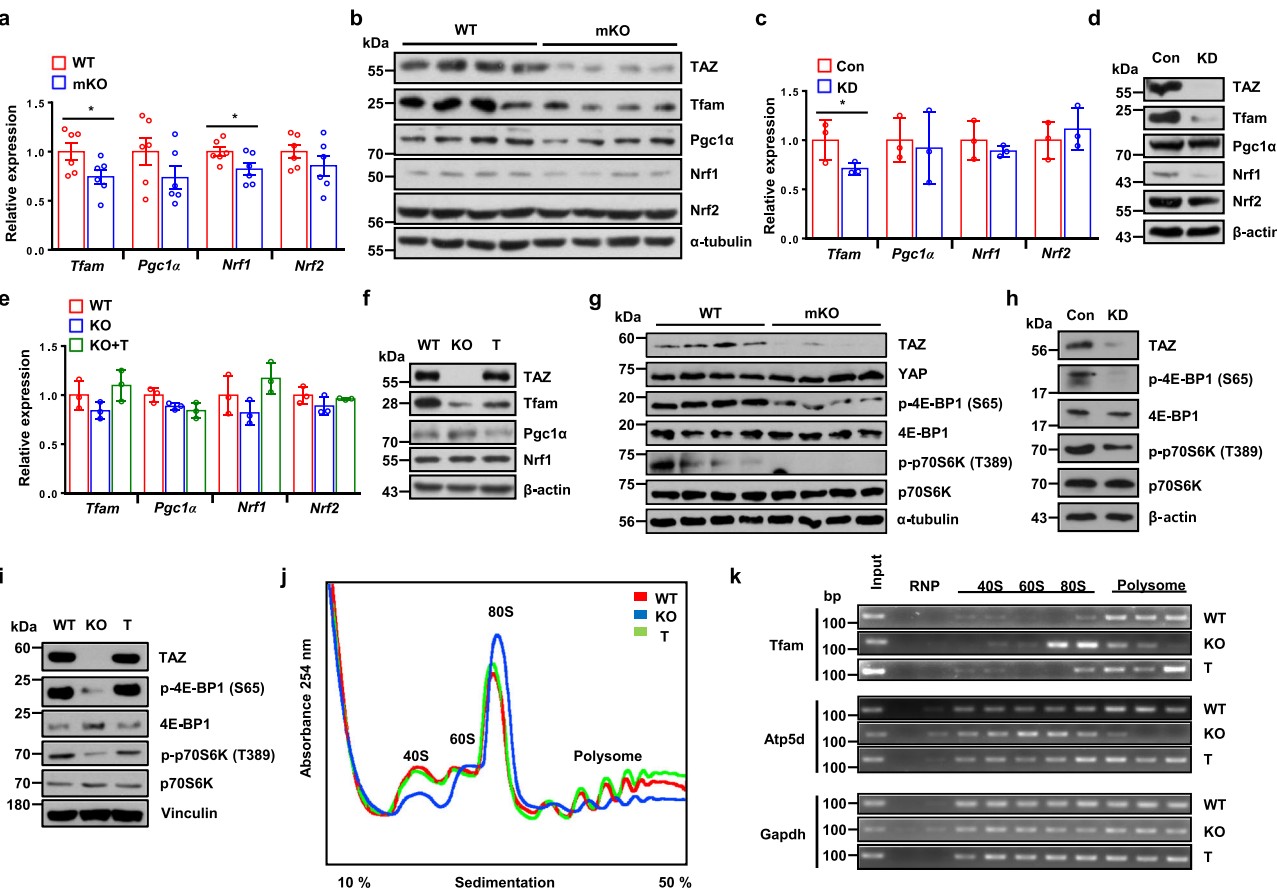

**Fig. 2 TAZ induces Tfam protein expression at the translational level. a** The expression of mitochondrial biogenic factors, including *Tfam*, *Ppargc1a* (*Pgc1α*), *Nrf1*, and *Nrf2*, was analysed using quantitative reverse transcription (qRT)-PCR in the gastrocnemius muscle of wild-type (WT) and muscle-specific TAZ knockout (mKO) mice; $n = 6$ for each condition (*Tfam*; *$p = 0.0498$, *Nrf1*; *$p = 0.0467$). **b** Protein was analysed via immunoblotting to detect TAZ and the indicated mitochondrial biogenic factors in (**a**). Alpha-tubulin was used as the loading control. **c** RNA was isolated from the control (Con) and TAZ knockdown (KD) C2C12 myotubes, and the expression of the indicated mitochondrial biogenic factors was analysed using qRT-PCR. The experiments were performed in triplicate (*Tfam*; *$p = 0.04$). **d** Protein from the Con and TAZ KD C2C12 myotubes was assessed via immunoblotting to detect TAZ and mitochondrial biogenic factors. Beta-actin was used as the loading control. **e** RNA was isolated from WT, TAZ-knockout (KO), and TAZ-rescued (T) mouse embryonic fibroblasts (MEF), and the indicated mitochondrial biogenic factors were analysed via qRT-PCR. **f** Protein from WT, KO, and T MEF was assessed via immunoblotting to detect TAZ and the indicated mitochondrial biogenic factors. Beta-actin was used as the loading control. **g** Protein was isolated from the gastrocnemius muscle of WT and mKO mice and analysed via immunoblotting to detect TAZ, YAP, and proteins involved in the translation of both phosphorylated and total forms. Alpha-tubulin was used as the loading control. **h** Protein obtained from the Con and TAZ KD C2C12 myotubes was assessed via immunoblotting to observe TAZ and proteins involved in translation. Beta-actin was used as the loading control. **i** Protein was isolated from WT, KO, and T MEF and analysed via immunoblotting to detect TAZ and proteins involved in translation. Vinculin was used as the loading control. **j** The polysome profile of cells described in panel **i** was evaluated continuously by measuring absorbance at 254 nm. The 40S, 60S, 80S, and polysome fractions are denoted by the corresponding peaks. **k** The distribution of *Tfam*, *Atp5d*, and *Gapdh* mRNA from cells described in (**j**) across a density gradient was analysed using reverse transcription (RT)-PCR. bp = base pairs. For **a**, **b**, **g** 8 to 10-week-old mice were used. Data are presented as mean ± SEM for (**a**), mean ± SD for (**c**, **e**). Statistical significance was analysed via two-sided *t*-test for (**a**) and one-sided *t*-test for (**c**). One-way ANOVA with Tukey's multiple comparison test was used for (**e**). Representative data was shown and experiments were performed at least twice with similar results for (**b**, **d**, **f–i**, **k**). Source data are provided as a Source data file.

plays an important role in translational processing. For direct examination of whether endogenous TAZ stimulates the translation of mitochondrial marker genes, polysomes of WT, TAZ KO, and TAZ-introduced TAZ KO MEF were sedimented in sucrose density gradients to separate efficiently and poorly translated mRNAs. As shown in Fig. 2j, TAZ KO cells exhibited decreased polysome content with a corresponding 80 S peak, suggesting that TAZ is required for complete polysome disassembly. These alterations were reversed by the introduction of TAZ into TAZ KO cells (Fig. 2j). We also assessed the effects of TAZ on the polysomal distribution of *Tfam* and *Atp5d* mRNAs. TAZ KO cells demonstrated a shift of these mRNAs toward lighter polysomes. However, the shift was reversed in

TAZ-introduced TAZ KO cells (Fig. 2k). These results demonstrate that TAZ plays an important role in the translation of mitochondrial genes, including *Tfam*.

**TAZ induces *Rheb* and *Rhebl1* expression to promote the initiation of the translation of *Tfam* mRNA.** Next, we assessed how endogenous TAZ stimulates the mTORC1 pathway and which of the TAZ target genes may be involved. Our previous ChIP-seq analysis data[21] indicated that Ras homolog enriched in brain like 1 (*Rhebl1*), an activator of mTORC1, is a TAZ target gene. We observed that the mRNA expression and protein levels of Rhebl1 and Rheb were decreased in TAZ mKO mice (Fig. 3a,

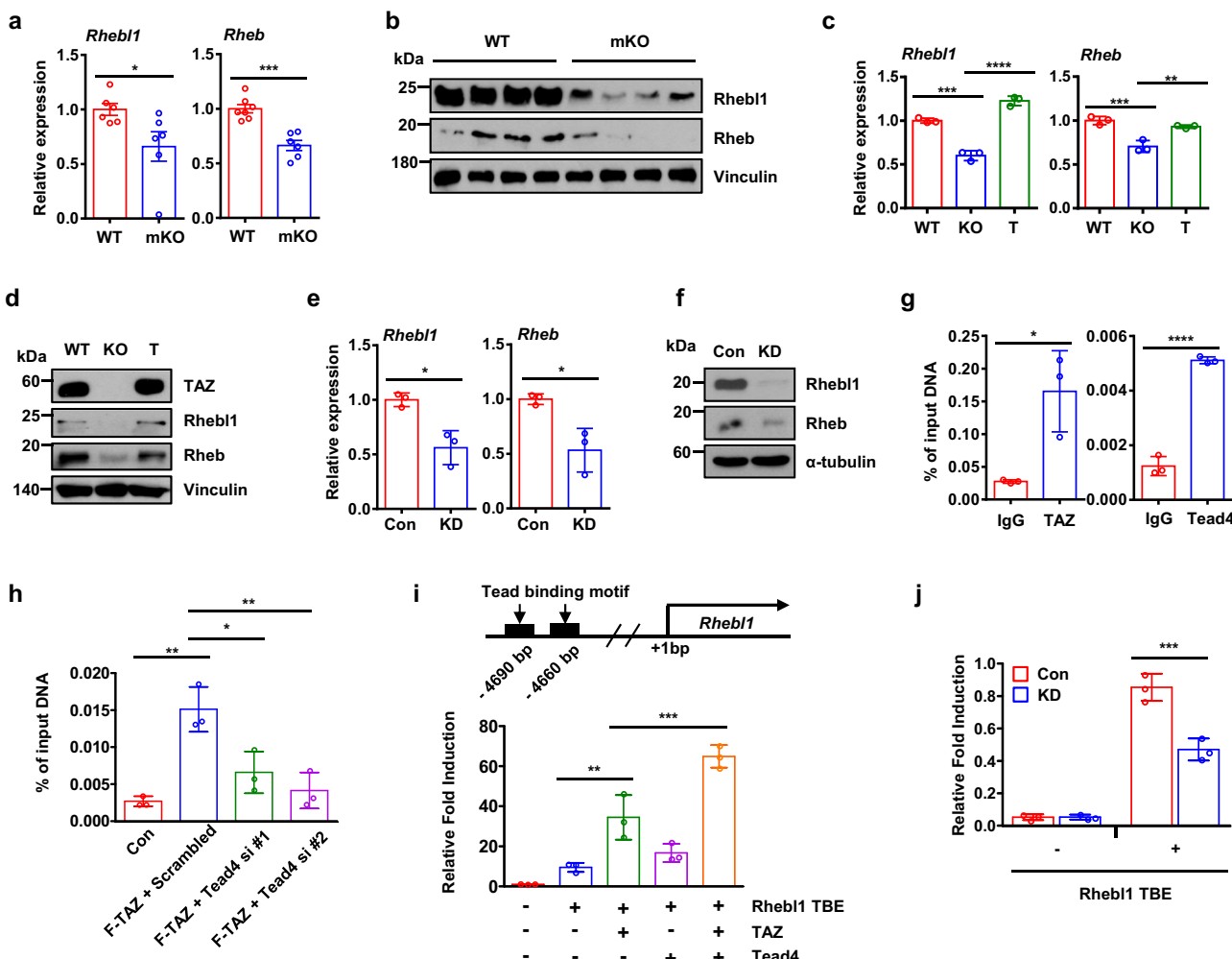

**Fig. 3 TAZ induces the expression of *Rheb* and *Rhebl1* to promote the initiation of the translation of *Tfam* mRNA. a** RNA was isolated from the gastrocnemius muscle of wild-type (WT) and muscle-specific TAZ knockout (mKO) mice, and the transcript levels of *Rheb* and *Rhebl1* were assessed using quantitative reverse transcription (qRT)-PCR; *n* = 6 (left), *n* = 7 for WT mice, and *n* = 6 for mKO mice (right) (*Rhebl1*; *p = 0.0426, *Rheb*; ***p = 0.0002). **b** Protein derived from the gastrocnemius muscle of WT and mKO mice was analysed via immunoblotting to determine the Rheb and Rhebl1 protein levels. Vinculin was used as the loading control. **c** The expression of *Rheb* and *Rhebl1* was assessed using qRT-PCR in WT, TAZ knockout (KO), and TAZ-rescued (T) mouse embryonic fibroblasts (MEF). The experiment was performed in triplicate (*Rhebl1*; ***p = 0.0001 for WT vs. KO, ****p < 0.0001 for KO vs. T, *Rheb*; ***p = 0.0007 for WT vs. KO, **p < 0.003 for KO vs. T). **d** Protein from WT, KO, and T MEF was analysed via immunoblotting to detect TAZ, Rheb, and Rhebl1. Vinculin was used as the loading control. **e** RNA was isolated from control (Con) and C2C12 myotubes with TAZ knockdown (KD). *Rhebl1* and *Rheb* expression was assessed using qRT-PCR. The experiment was performed in triplicate (*Rhebl1*; *p = 0.0105, *Rheb*; *p = 0.0171). **f** Protein from the Con and TAZ KD C2C12 myotubes was analysed via immunoblotting to detect the Rhebl1 and Rheb protein levels. Alpha-tubulin was used as the loading control. **g** Recruitment of TAZ (left) or TEAD4 (right) to the TAZ binding element (TBE) of the *Rhebl1* enhancer was verified using chromatin immunoprecipitation (ChIP)-quantitative PCR (qPCR). Sheared chromatin from the gastrocnemius muscle of WT mice was ChIPed with the anti-TAZ or anti-TEAD4 antibody. ChIPed DNA was analysed via qPCR using primers spanning the TBE region. Immunoglobulin G (IgG) was used as the ChIP control. The experiment was performed in triplicate (TAZ; *p = 0.0183, Tead4; ***p < 0.0001). **h** TEAD4 was depleted by two different siRNAs in FLAG-tagged TAZ (F-TAZ)-overexpressing C2C12 myoblasts, and ChIP was performed using an anti-FLAG antibody. ChIPed DNA was assessed via qPCR using a primer set covering the TBE region. Empty vector-transduced cells were used as ChIP controls. The experiment was performed in triplicate (Con vs. F-TAZ + Scrambled; **p = 0.0011, F-TAZ + Scrambled vs. F-TAZ + Tead4 si #1; *p = 0.0112, F-TAZ + Scrambled vs. F-TAZ + Tead4 si #2; **p = 0.0025). **i** Schematic description of *Rhebl1* cis-regulatory element regulated by TAZ-TEAD4. The transcriptional activity of the *Rhebl1* TBE was confirmed using a luciferase reporter gene assay. Approximately 500 bp of the genomic region covering the *Rhebl1* TBE were cloned into a pGL3 basic luciferase reporter plasmid together with the promoter region of *Rhebl1* (*Rhebl1* TBE-Luc). *Rhebl1* TBE-Luc was transfected into HEK293T cells with TAZ and/or TEAD4, and a reporter gene assay was performed 24 h after transfection. The experiment was performed in triplicate (*Rhebl1* TBE vs. *Rhebl1* TBE + TAZ; **p = 0.0033, *Rhebl1* TBE + TAZ vs. *Rhebl1* TBE + TAZ + Tead4; ***p = 0.0007). **j** Con and HEK293T cells with TAZ KD were transfected with *Rhebl1* TBE-Luc. A luciferase reporter assay was performed 24 h after transfection. The experiment was performed in triplicate (***p = 0.0002). For **a**, **b**, **g** 8 to 10-week-old mice were used. Data are presented as mean ± SEM for (**a**), and mean ± SD for (**c**, **e**, **g–j**). Statistical significance was analysed via two-sided *t*-test for (**a**, **e**, **g**) or one-way ANOVA with Tukey's multiple comparison test for (**c**, **h**, **i**). For (**j**) two-way ANOVA with Sidak's multiple comparison test was used. Representative data was shown and experiments were performed at least twice with similar results for (**b**, **d**, **f**). Source data are provided as a Source data file.

b). Similarly, Rhebl1 and Rheb mRNA expression and protein levels were also decreased in TAZ KO MEF (Fig. 3c, d) and TAZ KD C2C12 myotubes (Fig. 3e, f). In contrast, the introduction of TAZ into TAZ KO MEF restored the mRNA expression and protein levels corresponding to both Rhebl1 and Rheb (Fig. 3c, d). These results demonstrate that TAZ stimulates *Rhebl1* and *Rheb* expression to activate mTOR signalling. To study the mechanism by which endogenous TAZ stimulates *Rhebl1* expression, we searched for cis-regulatory elements in the *Rhebl1* promoter region. Previously, we analysed our ChIP-seq data to identify the genomic TAZ-binding region in FLAG-TAZ-expressing C2C12 myoblasts. The *Rhebl1* enhancer region (~4.6 kilobase pairs (kbp) upstream of the start site) contains a TAZ-binding element (TBE). We verified TAZ binding via ChIP-qPCR (Fig. 3g, left). Interestingly, this region also exhibited two TEAD binding sites and TEAD4 binding was also verified via ChIP analysis (Fig. 3g, right). In addition, TAZ recruitment to the TBE was abrogated by siRNA-mediated TEAD4 depletion (Fig. 3h). Next, we evaluated the transcriptional activity of the *Rhebl1* TBE region. Specifically, a fragment comprising approximately 500 bp of genomic DNA adjoining the *Rhebl1* TBE was cloned into a luciferase reporter

vector to synthesize the *Rhebl1* TBE-Luc plasmid (Fig. 3i). This plasmid was introduced into HEK293T cells with TAZ and/or TEAD4 expression plasmids, and luciferase activity was measured. As shown in Fig. 3i, TAZ and TEAD4 stimulated the activity of the reporter gene. Additionally, these plasmids were also introduced into the control and TAZ KD HEK293T cells and, as shown in Fig. 3j, luciferase activity was markedly decreased in TAZ KD cells. These results suggest that TAZ and TEAD4 are positive regulators of *Rhebl1* expression.

**Rhebl1 introduction recovers mitochondrial biogenesis through Tfam induction in mKO muscle.** To assess whether Rhebl1 is responsible for TAZ-mediated translational control, Rhebl1 expressing AAV6 virus was directly injected into mKO muscle tissue. As shown in Fig. 4a, *Rhebl1* expression was increased in mKO muscle 2 weeks after Rhebl1 viral introduction. In addition, muscle lysates were prepared and levels of mitochondrial proteins were analysed (Fig. 4b). Rhebl1 viral introduction increased Rhebl1 levels with induction of phosphorylated 4E-BP1, Tfam, Cox4, CytC, and Sdha (Fig. 4b). Increased

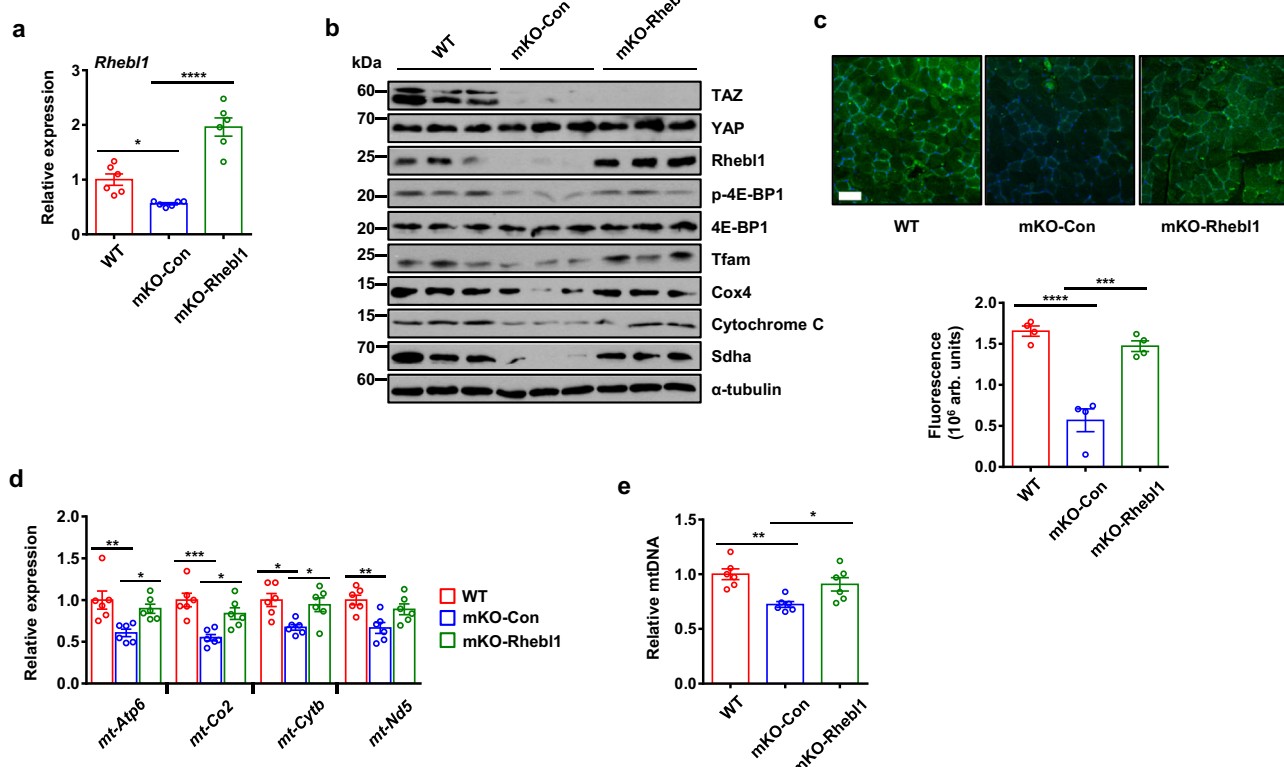

**Fig. 4 Rhebl1 re-introduction restored mitochondrial biogenesis in muscle-specific TAZ knockout mice. a** RNA was isolated from gastrocnemius muscle of wild type (WT) and muscle-specific TAZ knockout (mKO) mice intramuscularly transduced with pAAV6-CMV (mKO-Con) or pAAV6-Rhebl1 (mKO-Rhebl1) virus. Levels of *Rhebl1* transcripts were analysed by quantitative reverse transcription (qRT-PCR) to confirm Rhebl1 rescue. *n* = 6 for each condition (WT vs. mKO-Con; **p* = 0.0377, mKO-Con vs. mKO-Rhebl1; *****p* < 0.0001). **b** Protein extracted from gastrocnemius of mice in (**a**) was analysed by immunoblot assays to confirm levels of the indicated proteins. Alpha-tubulin was used as a loading control. Representative data was shown and experiment was performed twice with similar results. **c** Gastrocnemius muscle of mice in (**a**) was assessed by immunofluorescent staining for cytochrome c oxidase subunit 2 to visualise mitochondria. Cell nuclei were counterstained with DAPI. Scale bar = 50 μm. Fluorescence was quantified using ImageJ. *n* = 4 for each condition. (WT vs. mKO-Con; *****p* < 0.0001, mKO-Con vs. mKO-Rhebl1; ****p* = 0.0002). **d** RNA isolated from gastrocnemius muscle of mice described in (**a**) was assessed by qRT-PCR to determine transcript levels of mitochondria-encoded genes. *n* = 6 for each condition (*mt-Atp6*; ***p* = 0.0054 for WT vs. mKO-Con, **p* = 0.0378 for mKO-Con vs. mKO-Rhebl1, *mt-Co2*; ****p* = 0.0006 for WT vs. mKO-Con, **p* = 0.0185 for mKO-Con vs. mKO-Rhebl1, *mt-Cytb*; **p* = 0.0103 for WT vs. mKO-Con, **p* = 0.033 for mKO-Con vs. mKO-Rhebl1, *mt-Nd5*; ***p* = 0.0049 for WT vs. mKO-Con). **e** Genomic DNA isolated from gastrocnemius muscle of mice described in (**a**) was analysed by quantitative PCR for relative quantification of mitochondrial DNA copy number. *n* = 6 for each condition (WT vs. mKO-Con; ***p* = 0.0026, mKO-Con vs. mKO-Rhebl1; **p* = 0.039). For (**a**, **c**–**e**), data are shown as means ± SEM. Statistical significance was analysed via one-way ANOVA with Tukey's multiple comparison test. Source data are provided as a Source data file.

expression of cytochrome c oxidase subunit 2 was further verified in Rhebl1 virus-transduced muscle by immunofluorescent staining (Fig. 4c). Mitochondrial genes including mitochondrially encoded ATP synthase 6 (mt-Atp6), mitochondrially encoded cytochrome c oxidase II (mt-Co2), mitochondrially encoded cytochrome b (mt-Cytb), and mitochondrially encoded NADH:ubiquinone oxidoreductase core subunit 5 (mt-Nd5) were also induced after Rhebl1 introduction (Fig. 4d). Finally, Rhebl1 introduction in mKO muscle increased mitochondrial DNA content (Fig. 4e).

To study further TAZ-Rhebl1 axis in vitro, Rhebl1 was reintroduced into TAZ KO MEF (Supplementary Fig. 3a), which restored the activation of 4E-BP1 and subsequent induction of mitochondrial proteins, including Tfam (Supplementary Fig. 3a). In addition, the expression of mitochondrial-encoded genes including mt-Atp6, mt-Co2, mt-Cytb, and mt-Nd5 (Supplementary Fig. 3b) along with the mitochondrial DNA copy number (Supplementary Fig. 3c), and ATP production (Supplementary Fig. 3d) were restored in Rhebl1-rescued KO MEF. These results suggest that Rhebl1 plays a major role in TAZ-mediated mitochondrial biogenesis and TAZ stimulates Rhebl1 expression to induce mitochondrial biogenesis.

Furthermore, mTOR activator MHY1485 was administered to mKO mice to study the TAZ-mTOR axis. This treatment increased levels of phosphorylated 4E-BP1 and p70S6K proteins with induction of Tfam, Cox4, and cytochrome C (Supplementary Fig. 4a). Cox4 induction was further verified by immunofluorescent staining (Supplementary Fig. 4b). Mitochondrial genes including mt-Atp6, mt-Co2, mt-Cytb, and mt-Nd5 were also induced after treatment with mTOR activator (Supplementary Fig. 4c). Finally, mTOR activator administration in mKO muscle increased mitochondrial DNA content (Supplementary Fig. 4d). Thus, these results suggested that the TAZ-Rhebl1-mTOR axis plays an important role in mitochondrial biogenesis.

**TAZ stimulates exercise-induced mitochondrial biogenesis and increases respiratory metabolism and activity.** Since exercise stimulates mitochondrial biogenesis[26–28], we investigated whether exercise-induced mitochondrial biogenesis is mediated by TAZ activation. To evaluate the effect of TAZ on exercise-induced mitochondrial biogenesis, WT and mKO mice were subjected to endurance exercise training, and markers of mitochondrial biogenesis were then analysed. The expression of Cox4, which is a mitochondrial marker protein, was increased in the trained WT mice, but not in trained mKO mice as shown by immunofluorescent staining (Fig. 5a). Mitochondrial DNA copy number was also increased in trained WT mice, but the increase was not observed in mKO mice (Fig. 5b). In addition, the level of mitochondrial marker proteins including Tfam, Sdha, Vdac, Cox4, and Cytochrome C was increased in trained WT mice but not in mKO mice (Fig. 5c). Finally, the transcript level of mitochondrial-encoded genes including mt-Atp6, mt-Co2, mt-Cytb, and mt-Nd5, which are stimulated by Tfam transcription factor, was increased in trained WT mice but not in trained mKO mice (Fig. 5d). Notably, the transcript level of nuclear-encoded mitochondrial genes including ATP synthase, H+ transporting, mitochondrial F1 complex, alpha subunit 1 (Atp5a1), isocitrate dehydrogenase 3 (NAD+), gamma (Idh3g), NADH:ubiquinone oxidoreductase subunit A10 (Ndufa10), and ubiquinol-cytochrome c reductase core protein 1 (Uqcrc1), which is regulated by PGC1α, was induced in both trained WT and mKO mice (Fig. 5e), suggesting that PGC1α activity is not altered in mKO mice. Indeed, PGC1α level was increased in both trained WT and mKO mice (Fig. 5c). We also observed that trained WT mice muscle demonstrated an increased expression of Rhebl1, phospho-4E-BP1, and

phospho-p70S6K, indicating that trained muscle has increased translational potential (Fig. 5c). However, the increase was not significant in mKO mice (Fig. 5c). These results suggest that TAZ stimulates exercise-induced mitochondrial biogenesis via the translational induction of Tfam.

We next evaluated the role of TAZ in exercise-induced changes with respect to metabolic parameters. Exercise-trained WT mice demonstrated increased $O_2$ consumption, $CO_2$ production, and energy expenditure compared to control WT mice in the dark. However, these changes were reduced in the mKO mice (Fig. 5f).

To further evaluate the role of TAZ in mitochondrial function after exercise, muscle mitochondria were isolated from untrained or trained WT and mKO mice and their respiration activity was analysed using XF analyser (Supplementary Fig. 5a, b). After exercise, trained WT mitochondria exhibited increased basal activity, but this increase was not significant in trained mKO mitochondria. Similarly, trained WT mitochondria showed significantly increased maximal respiratory capacity after treatment with FCCP compare to trained mKO mitochondria. Taken together, these results demonstrated that TAZ is a novel regulator of mitochondrial biogenesis in response to extracellular signals, including exercise.

## Discussion

In this study, we demonstrated that TAZ stimulates mitochondrial biogenesis and exercise-induced muscle adaptation. Mechanistically, TAZ induces Rheb/Rhebl1 to inactivate 4E-BP1, thereby stimulating Tfam translation for mitochondrial-encoded gene expression (Fig. 6). PGC1α, a master regulator of mitochondrial biogenesis, stimulates Tfam expression leading to the induction of mitochondrial gene transcription[7,8]. In this study, the results demonstrated that PGC1α levels and activity were not altered in WT and TAZ mKO mice muscle. However, Tfam level was significantly decreased in mKO mice muscle. Physiologically, PGC1α target genes, such as nuclear-encoded mitochondrial genes including Tfam were induced in the exercise-trained WT and mKO mice. However, Tfam protein was not induced in mKO mice after exercise. Therefore, we demonstrated that TAZ is a novel stimulator of mitochondrial biogenesis, which is mechanistically different from PGC1α, and that both PGC1α and TAZ are important for Tfam production via transcription and translation, respectively. Therefore, they enhance mitochondrial biogenesis in response to exercise.

In addition to decreased mitochondrial-encoded mitochondrial proteins, TAZ knockout also decreased the level of certain nuclear-encoded mitochondrial proteins including Sdha and Cox4 (Fig. 1), suggesting that there is another role of TAZ in the regulation of nuclear-encoded mitochondrial genes. As shown in Fig. 2k, translation of Atp5d mRNA, which is a nuclear-encoded mitochondrial gene, was retarded in TAZ KO MEF. Similar results were reported by the Sonenberg group, demonstrating translational downregulation of many nuclear-encoded mitochondrial genes by mTOR inhibition through genome-wide polysome profiling[24,29]. Thus, these results suggested that the TAZ-Rheb/Rhebl1-mTOR axis plays important roles in the translational regulation of mitochondrial genes.

mTORC1 is activated after resistance and not endurance exercise owing to muscle hypertrophy[26,30]. This observation raised a question whether mTORC1 activity is induced during endurance exercise training in our study. Therefore, we investigated the mTORC1 activity and observed that during the resting/feeding period after single bout of endurance exercise, mTORC1 activity was increased (Supplementary Fig. 6). These results suggest that the resting/feeding period after endurance exercise is

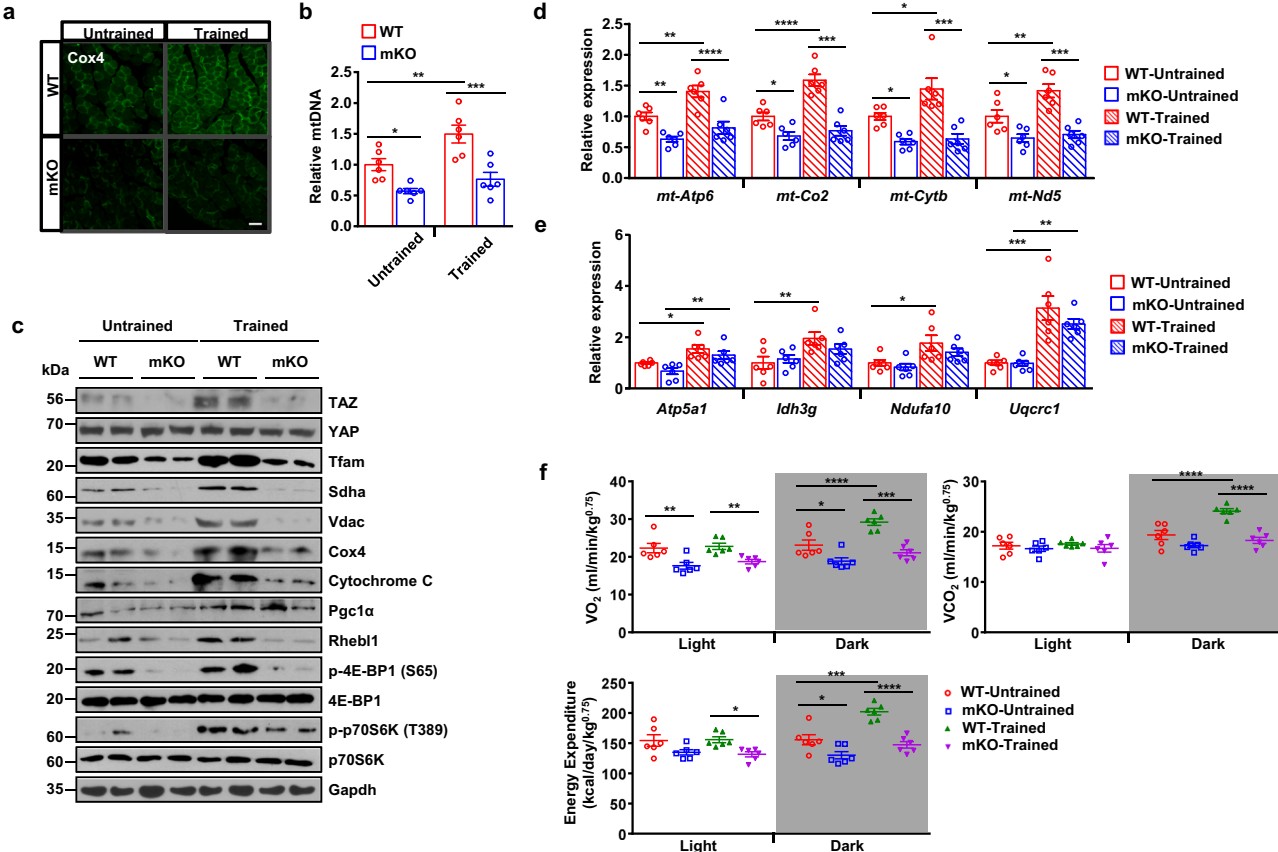

**Fig. 5 TAZ stimulates exercise-induced mitochondrial biogenesis and increases respiratory metabolism and activity. a** Wild-type (WT) and muscle-specific TAZ knockout (mKO) mice were subjected to endurance exercise training. The gastrocnemius muscle obtained from these mice was isolated and embedded in paraffin. The sectioned tissues were immunostained with an anti-COX4 antibody to visualise the mitochondria. Scale bar = 50 μm. **b** Genomic DNA was isolated from the gastrocnemius muscle of mice described in (**a**). Relative mitochondrial DNA copy number was determined via quantitative PCR using primers for mitochondrial-encoded *Cox2* and nuclear-encoded *β-globin*. The Ct values of mitochondrial *Cox2* were normalized to those of nuclear *β-globin*; $n = 6$ for each genotype (WT untrained vs. mKO untrained; $*p = 0.0188$, WT untrained vs. WT trained; $**p = 0.0069$, WT trained vs. mKO trained; $***p = 0.0002$). **c** Protein from the gastrocnemius muscle of mice described in (**a**) was assessed via immunoblotting to observe the indicated proteins. GAPDH was used as the loading control. **d** RNA was isolated from the gastrocnemius muscle of mice described in (**a**). Mitochondrial-encoded marker gene expression was analysed via quantitative reverse transcription (qRT)-PCR, $n = 6$ for each genotype (*mt-Atp6*; $**p = 0.0035$ for WT untrained vs. mKO untrained, $**p = 0.0071$ for WT untrained vs. WT trained, $****p < 0.0001$ for WT trained vs. mKO trained, *mt-Co2*; $*p = 0.0174$ for WT untrained vs. mKO untrained, $****p < 0.0001$ for WT untrained vs. WT trained, $****p < 0.0001$ for WT trained vs. mKO trained, *mt-Cytb*; $*p = 0.0214$ for WT untrained vs. mKO untrained, $*p = 0.0124$ for WT untrained vs. WT trained, $****p < 0.0001$ for WT trained vs. mKO trained, *mt-Nd5*; $*p = 0.0186$ for WT untrained vs. mKO untrained, $**p = 0.0059$ for WT untrained vs. WT trained, $****p < 0.0001$ for WT trained vs. mKO trained). **e** Nuclear-encoded mitochondrial gene expression in the gastrocnemius muscle of mice in (**a**) was analysed via qRT-PCR, $n = 6$ for each genotype (*Atp5a1*; $*p = 0.0137$ for WT untrained vs. WT trained, $**p = 0.0048$ for eKO untrained vs. eKO trained, *Idh3g*; $**p = 0.0089$ for WT untrained vs. WT trained, *Ndufa10*; $*p = 0.0205$ for WT untrained vs. WT trained, *Uqcrc1*; $****p < 0.0001$ for WT untrained vs. WT trained, $**p = 0.0011$ for mKO untrained vs. mKO trained). **f** The respiratory metabolism of mice described in (**a**) was analysed using the Oxylet system. Data are shown as the average values observed under light and dark conditions; $n = 6$ for each condition (VO$_2$-light; $**p = 0.0033$ for WT untrained vs. mKO untrained, $**p = 0.0098$ for WT trained vs. mKO trained, VO$_2$-dark; $*p = 0.0146$ for WT untrained vs. mKO untrained, $***p < 0.0007$ for WT untrained vs. WT trained, $****p < 0.0001$ for WT trained vs. mKO trained, VCO$_2$-dark; $****p < 0.0001$ for WT untrained vs. WT trained, $****p < 0.0001$ for WT trained vs. mKO trained, Energy expenditure-light; $*p = 0.0234$ for mKO untrained vs. mKO trained, Energy expenditure-dark; $*p = 0.021$ for WT untrained vs. mKO untrained, $***p = 0.0001$ for WT untrained vs. WT trained, $****p < 0.0001$ for WT trained vs. mKO trained). Ten to twelve-week-old mice were used for all panels. Data are presented as mean ± SEM for (**b**, **d**–**f**). Statistical significance was analysed via two-way ANOVA with Sidak's multiple comparison test. Representative data was shown and experiments were performed at least twice with similar results for (**a**, **c**). Source data are provided as a Source data file.

important for translational activation of mitochondrial genes. Therefore, our results demonstrate that mitochondrial biogenesis induced by endurance exercise training is occurred by both PGC1α-mediated *Tfam* expression during exercise and TAZ-mediated Tfam translation after exercise (Fig. 6).

Protein synthesis for exercise-mediated mitochondrial biogenesis has been examined previously. Translational activation owing to exercise predominantly supports phenotypic gains in muscle mitochondrial function and hypertrophy at all ages[31].

Furthermore, extensive remodelling of the mitochondrial proteome is observed after just 7 days of exercise training, with enhanced ATP-generating capacity[32]. Additionally, differences have been observed between the proteomic profiles of the resting muscles of endurance exercise-trained and untrained individuals, as well as in proteome modulation in response to acute exercise[33]. Together with previous reports, our results suggest that TAZ is an important signalling mediator for exercise-induced muscle protein synthesis.

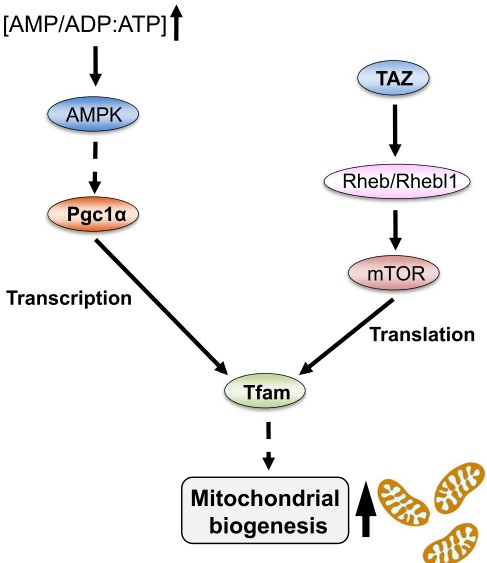

Exercise Training

**Fig. 6 Summarized model TAZ protein was stabilized and activated by exercise training.** Activated TAZ translocates into the nucleus and induces *Rheb/Rhebl1* expression by binding to the TEAD transcription factor. During a resting period, elevated Rheb/Rhebl1 levels amplify TORC1 signalling for upregulating the translation of *Tfam* mRNA. Increased Tfam protein enhances mitochondrial biogenesis by the upregulation of mitochondrial DNA replication and mitochondria-encoded gene expression.

Collectively, our results demonstrate that TAZ induces exercise-mediated muscle adaptations to meet energy demands, including the stimulation of mitochondrial biogenesis and cellular respiration rates. Our results suggest a novel molecular mechanism for mitochondrial biogenesis via TAZ and exercise-induced muscle adaptation.

## Methods

**Mice.** Muscle-specific TAZ knockout mice were previously described[21]. In brief, to develop muscle-specific TAZ knockout mice, a floxed allele containing LoxP sites flanking the *Taz* exon 2 region was constructed. Mice containing the floxed *Taz* allele were crossed with *MCK-Cre* mice for muscle-specific *Taz* knockout. The floxed exon 2 of *Taz* was deleted by Cre-mediated recombination in the presence of the *MCK-Cre* allele. *MCK-Cre* [FVB-Tg (Ckmm-cre) 5Khn/J, #006405] mice were purchased from the Jackson Laboratory (Bar Harbor, ME, USA). The procedures for animal care and experimentation were approved by the Institutional Animal Care and Use Committee of Korea University (KUIACUC-2019-0093) and were in compliance with the institutional guidelines. Mice were housed at 22 °C, 12-h light/12-h dark and 50% humidity.

**Exercise training.** Ten to twelve-week old wild-type and muscle-specific TAZ knockout (mKO) mice were trained using the following exercise program. At all exercise bouts, mice were first run at 10 m/min for 5 min. The slope angle of treadmill was set to 5°. On the first day of the first week, exercise was performed at 15 m/min for 20 min. The duration of the exercise was increased to 60 min by 10 min per day. In the second week, the mice were exercised at 15 m/min for 60 min. In the third week, running was performed at 17 m/min for 60 min. Finally, in the fourth week, mice were run at 19 m/min for 60 min. Exercise was performed 5 days per week for the first 3 weeks, and for 2 days in the last week. Mice were sacrificed 24 h after last exercise bout. During resting period, mice were fed with normal chow diet *ad libitum*.

**Acute exercise.** Ten to twelve-week old wild-type and muscle-specific TAZ knockout (mKO) mice were subject to run treadmill at 15 m/min for 1 h. The slope angle of treadmill was set to 5°. Mice were sacrificed 0, 3, and 24 h after exercise bout for further analysis. During resting period, mice were fed with normal chow diet *ad libitum*

**Cell culture.** Mouse embryonic fibroblasts (MEF) were isolated from 12.5- to 13.5-days-post-coitum mouse embryos. The MEF were maintained in Dulbecco's modified Eagle's medium (DMEM) supplemented with 10% fetal bovine serum (FBS) and penicillin/streptomycin. C2C12 myoblasts were maintained in DMEM supplemented with 20% FBS and penicillin/streptomycin. For myogenic differentiation, C2C12 myoblasts were seeded at a density of $6 \times 10^4$ cells/cm$^2$ and the medium was changed to 2% horse serum-containing DMEM the following day. In all cases, the cells were incubated at 37 °C in a 5% CO$_2$ atmosphere.

**Antibodies.** Antibodies against Cytochrome C (#4280), Cox4 (#4850), Sdha (#5839), Vdac (#4661), Hsp60 (#4870), Rheb (#13879), p-4E-BP1 (#9451), 4E-BP1 (#9452), p-p70S6K (#9234), p70S6K (#2708), p-AMPKα (#2535), AMPKα (#2603), Vinculin (#13901), TAZ/YAP (#8418), and YAP (#4912) were purchased from Cell Signaling Technology (Danvers, MA, USA). Antibodies against Rhebl1 (sc-514095), α-Tubulin (sc-5286), PGC1α (sc-13067), and GAPDH (sc-166574) were purchased from Santa Cruz Biotechnology (Dallas, TX, USA). Anti-Tead4 (ab58310), anti-cytochrome c oxidase subunit 2 (ab198286), anti-Nrf1 (ab34682) and -Nrf2 (ab88746, also known as GABPB2) antibodies were purchased from Abcam (Cambridge, UK). Antibodies against FLAG (F1804) and β-Actin (A5441) were purchased from Sigma-Aldrich (St. Louis, MO, USA) and the anti-Tfam antibody (ARP36992-P050) from Aviva Systems Biology (San Diego, CA, USA). Anti-TAZ (NB110-58359) was purchased from Novus Biologicals (Centennial, CO, USA). Fluorophore-conjugated secondary antibody (A11008) was purchased from Thermo Fisher Scientific (Waltham, MA, USA). HRP-conjugated secondary antibodies (ADI-SAB-300-J and ADI-SAB-100-J) were purchased from Enzo Life Sciences (Farmingdale, NY, USA). For immunoblot assays, antibodies were diluted to working concentrations of 1:200 (Rhebl1), 1:1000 (PGC1α, Nrf1, Nrf2, Tfam, p-4E-BP1, 4E-BP1, p-p70S6K, p70S6K, p-AMPKα, and AMPKα), 1:2000 (TAZ/YAP, YAP, Rheb, Vinculin, GAPDH, and α-Tubulin), or 1:5000 (Cytochrome C, Cox4, Sdha, Vdac, Hsp60, FLAG, and β-Actin). For immunostaining, Cox4 antibody was diluted to 1:500 and cytochrome c oxidase subunit 2 antibody was diluted to 1:50. For chromatin immunoprecipitation, TAZ (NB110-58359), FLAG and Tead4 antibodies were diluted to 0.5 μg of antibody per μg of DNA.

**Cloning of the luciferase construct for the *Rhebl1* TBE.** For *Rhebl1* TBE luciferase reporter gene constructs, the promoter region of the *Rhebl1* gene, which covers approximately 500 bp upstream of the transcription start site, was amplified using mouse genomic DNA as a template and inserted into the *Xho*I/*Hind*III restriction sites of the pGL3 basic vector (Promega, Madison, WI, USA). A DNA segment covering approximately 500 bp of the *Rhebl1* TBE-containing region was amplified and cloned into the *Kpn*I/*Nhe*I sites of the pGL3 basic vector. The primers used are listed in Supplementary Table 1.

**Viral constructs for gene knockdown and overexpression.** Retroviral constructs for Rhebl1 expression (pBabe puro-Rhebl1) were constructed by Gibson assembly. Specifically, the pBabe-puro vector was linearized by *Sna*BI digestion and mouse *Rhebl1* coding sequences were amplified by Phusion polymerase (F530L, Thermo Fisher Scientific, Waltham, MA, USA) and pCMV6-Rhebl1 template (MR215936, Origene, Rockville, MD, USA). The amplified products were assembled with the linearized vector using Gibson assembly (NEBuilder High-Fidelity DNA Assembly Master Mix, E2621L, New England Biolabs, Ipswich, MA, USA). PCR primer sequences for amplification of the mouse *Rhebl1* coding region were: Forward: 5′-GGATCCCAGTGTGGTGGTACGTCGACTGGATCCGGTAC-3′ and Reverse: 5′-CTGTGCTGGCGAATTCCTACTTAAACCTTATCGTCGTCATC-3′. Vectors of adeno-associated virus (AAV) for Rhebl1 expression (AAV6-Rhebl1) and TAZ knockdown (AAV6-shTAZ) were constructed by Gibson assembly as described above. pAAV-CMV (Cat. #6651, Takara Bio, Kusatsu, Shiga, Japan) was linearized by EcoRI (FD0274, Thermo Fisher Scientific). Murine *Rhebl1* coding region and hairpin sequence for mouse TAZ knockdown were amplified by Phusion polymerase (F530L, Thermo Fisher Scientific). pCMV6-Rhebl1 (MR215936, Origene) and pSRP-mTAZ (#31795, Addgene, Watertown, MA, USA) was used as a template for AAV6-Rhebl1 and AAV6-shTAZ, respectively. Primer sequences for AAV6-Rhebl1 were: Forward: 5′-AAAGAATTGGGATTCGCGAGGTCGACTGG ATCCGGTAC-3′ and Reverse: 5′-CACTAGTGTCGACTCTAGAGTTAAACCT TATCGTCGTCATCC-3′. Primer sequences for AAV6-shTAZ were: Forward: 5′-AAAGAATTGGGATTCGCGAGGAACGCTGACGTCATCAAC-3′ and Reverse: 5′-CACTAGTGTCGACTCTAGAGTCGAGTTCCAAAAAGATGAAT C-3′. Vectors of AAV for WT and mutant TAZ expression (AAV6-mTAZ and AAV6-mTAZS51A) were constructed by subcloning of TAZ coding region from pcDNA3.1-mTAZ FLAG or pcDNA3.1-mTAZ (S51A) FLAG.

**Retroviral transduction for stable cell lines.** Viral vectors were transfected into Phoenix cells with virus particle-packaging vectors. After 24 h, the virus-containing medium was collected and filtered through a 0.45-μm filter. The viral supernatant was then added to the target cells with 4 μg/μL of polybrene. To select infected target cells, 4 μg/mL (for C2C12) or 1 μg/mL (for mouse embryonic fibroblasts) of puromycin was added to the growth medium for 1–2 weeks. The pSRP-mTAZ plasmid (#31795, Addgene, Watertown, MA, USA) was used for TAZ knockdown.

The pBabe puro-mTAZ plasmid (#31791, Addgene, Watertown, MA, USA) was used for TAZ overexpression. The pBabe puro-Rhebl1 was used for Rhebl1 rescue.

**Adeno-associated virus production and local transduction into mouse muscle**. AAVpro 293T (Cat. #632273, Takara Bio, Kusatsu, Shiga, Japan) cells were seeded on a 150 mm dish at a density of $6 \times 10^4$ cells/cm$^2$ and transfected with AAV plasmid constructs using TransIT-VirusGEN® Transfection Reagent (MIR 6700, Mirus Bio, Madison, WI, USA). 12 h after transfection, the culture medium was changed to 2% FBS-containing DMEM. 3 days after transfection, 0.5 M EDTA (pH 8.0) was added to the culture dish and incubated at room temperature for 10 min. Then, cells were collected and centrifuged at $1750 \times g$ at 4 °C for 10 min. The supernatant was removed and AAV particles were isolated from cell pellets using AAV Extraction Solution (Cat. #6666, Takara Bio). Subsequent steps followed the manufacturer's protocol. After isolation, AAV viral titer was determined using AAVpro® Titration Kit (Cat. #6233, Takara Bio). For local transduction of gastrocnemius muscle, 50 µl containing $5 \times 10^{10}$ vector genomes of AAV virus was administered by intramuscular injection. Virus was administered at 4 points of muscle tissue. AAV6-CMV virus was administered as a control. Mice were anesthetized by isoflurane during injection and sacrificed 2 weeks after virus administration. For rescue of WT TAZ and mutant TAZ, AAV6-TAZ or AAV6-TAZ S51A virus was administered 4 days after AAV6-shTAZ transduction. AAV6-CMV virus was used as a control.

**mTOR activator administration into mice**. mTOR activator MHY1485 was purchased from ChemScene (CS-3946, Monmouth Junction, NJ, USA). 10% DMSO and 90% corn oil were sequentially added to MHY1485 for dissolution. MHY1485 was administered at 10 mg/g body weight via intraperitoneal injection. Injection was performed every 2 days for 2 weeks.

**siRNA-mediated gene knockdown**. C2C12 myoblasts were plated on cell culture dishes at a density of $1 \times 10^4$ cells/cm$^2$. After 24 h, the cells were transfected with siRNAs for the target genes using Lipofectamine 2000 (Invitrogen, Carlsbad, CA, USA) and incubated for 48 h. The cells were then harvested for further analysis. The target sequences for RNA interference are listed in Supplementary Table 1.

**Immunoblotting**. Cells were harvested and lysed in TNE lysis buffer (20 mM Tris-HCl [pH 7.5], 1% NP-40, 150 mM NaCl, 2 mM EDTA [pH 8.0], 50 mM NaF, 1 mM Na$_3$VO$_4$) supplemented with the appropriate protease inhibitors. For tissue samples, the dissected tissue was homogenized in pre-chilled RIPA buffer (150 mM NaCl, 50 mM Tris-HCl [pH 7.4], 1 mM EDTA, 1% NP-40, 0.5% sodium deoxycholate, 0.1% SDS, 1 mM Na$_3$VO$_4$, 1 mM NaF) with protease inhibitors using a tissue grinder. Total protein was quantified by the Bradford assay. Samples containing equal amounts of total protein were denatured by boiling for 5 min in 4× sodium dodecyl sulfate (SDS) sample buffer. To prepare phospho-proteins, 1× SDS sample buffer was directly applied to the cells. The cells were harvested, sonicated, and boiled at 98 °C for 5 min. SDS-polyacrylamide gel electrophoresis (PAGE) was performed using a Bio-Rad gel running system (Bio-rad, Hercules, CA, USA). Separated proteins on the gel were transferred to polyvinylidene difluoride (PVDF) membranes. For blocking, 5% non-fat dry milk in Tris-buffered saline with Tween 20 (TBST) was used. Membranes were incubated with specific primary antibodies overnight at 4 °C, followed by incubation with horseradish peroxidase (HRP)-conjugated secondary antibody for 1 h. After three washes with TBST for 5 min, the immunoblots were visualised using an enhanced chemiluminescence (ECL) system.

**Chromatin immunoprecipitation (ChIP)-quantitative PCR (qPCR)**. Control and FLAG-tagged TAZ-expressing C2C12 myoblasts were used for ChIP. Cells ($4 \times 10^6$) were seeded on a 150 cm$^2$ culture dish and incubated in growth medium for 24 h. The cells were then fixed in 0.75% formaldehyde to cross-link proteins and genomic DNA, and 125 mM glycine was added for quenching. Cells were then washed with ice-cold phosphate-buffered saline (PBS) and harvested. Harvested cells were lysed with FA lysis buffer (50 mM HEPES-KOH [pH 7.5], 140 mM NaCl, 1 mM EDTA [pH 8.0], 1% Triton X-100, 0.1% sodium deoxycholate, 0.1% SDS, protease inhibitor cocktail) and the chromatin was sheared by sonication to obtain 500–1000 bp fragments for ChIP-qPCR. In case of ChIP with muscle tissues, mouse gastrocnemius muscle was isolated and chopped by using curved surgical scissors on ice. Tissues were immersed in 1.5% formaldehyde containing ice-cold PBS and rotated at room temperature (RT) for 15 min. 0.125 M of glycine was added and samples were rotated at RT for 5 min to stop cross-linking. After 5 min centrifugation at $40 \times g$, supernatant was discarded and tissues were washed with ice-cold PBS. Then, tissues were homogenized in ice-cold PBS by tissue grinder. After checking unicellular suspension by microscope, cells were centrifuged for 10 min at $75 \times g$. FA lysis buffer was added to cell pellets and sonicated as described above. Sonicated samples were centrifuged to pellet the cell debris and the supernatant was collected as the chromatin preparation. The DNA concentration of the chromatin preparation was determined and input samples were removed. For chromatin capture, TAZ, FLAG, or TEAD4 antibodies were added to the samples and incubated at 4 °C for 16 h. Then, protein G beads (P-3296, Sigma Aldrich, St. Louis, MO, USA) coated with salmon sperm DNA and BSA were added to the antibody-bound samples and incubated at 4 °C for 6 h. The samples were then washed three times with wash buffer (0.1% SDS, 1% Triton X-100, 2 mM EDTA [pH 8.0], 150 mM NaCl, 20 mM Tris-HCl [pH 8.0]) and washed with final wash buffer (0.1% SDS, 1% Triton X-100, 2 mM EDTA [pH 8.0], 500 mM NaCl, 20 mM Tris-HCl [pH 8.0]). For elution, the samples were incubated with elution buffer (1% SDS, 100 mM NaHCO$_3$) at 30 °C for 15 min with rotation. RNase A was then added to the samples, which were incubated at 65 °C for 6 h to reverse the cross-linked chromatin. Samples were finally purified by gel extraction/PCR purification kit (Thermo Fisher Scientific, Waltham, MA, USA) and analysed by ChIP-qPCR. Light Cycler 480 Software (v1.5.0 SP4) was used to analyse ChIP-qPCR data.

**Bioinformatics analysis of ChIP-sequencing data**. Information for ChIP-sequencing peaks was obtained from a public library (ArrayExpress, E-MTAB-6764). Motif analysis of the *Rhebl1* TBE region was performed by JASPAR.

**Genomic DNA isolation and mitochondrial DNA quantification**. Growing cells were harvested in 1× PBS and centrifuged to obtain cell pellets. For isolation from muscle, muscle tissues were collected from mouse gastrocnemius muscle and washed with ice-cold PBS. The cell pellets or tissues were lysed in lysis buffer (100 mM Tris-HCl [pH 8.5], 5 mM EDTA [pH 8.0], 200 mM NaCl, 0.2% [w/v] SDS, 100 µg/mL proteinase K) at 55 °C for 16 h with frequent agitation. The digested samples were vigorously shaken and centrifuged. The supernatant was then mixed with isopropanol at room temperature. The genomic DNA precipitate was extracted using a pipette tip, dried, and dissolved in Tris-EDTA buffer (pH 8.0). The genomic DNA was analysed by quantitative real-time PCR (LC480, Roche, Basel, Switzerland). The relative mitochondrial DNA copy number was determined by the ratio of the Ct value of the mitochondrial *Cox2* (*mt-Co2*) gene to that of the nuclear *β-globin* gene calculated from the following formula: amplification efficiency ^ (Ct reference – Ct target). Light Cycler 480 Software (v1.5.0 SP4) was used to analyse data.

**Gene expression analysis**. Total RNA was isolated from cells or tissues using TRIzol reagent (Invitrogen, Carlsbad, CA, USA). Then, cDNA was synthesized using M-MLV reverse transcriptase and the transcript level was quantified using the LightCylcler480 system (Roche, Basel, Switzerland) with target gene-specific primers. In all experiments, *Gapdh* expression was assessed as a reference value for target gene expression. To obtain the target-to-reference gene ratio, the following formula was used: amplification efficiency ^ (Ct reference – Ct target). Calculated ratios were normalized to those of the control condition and presented as relative fold induction. Light Cycler 480 Software (v1.5.0 SP4) was used to analyse data.

**Polysome profiling**. Wild-type (WT) and TAZ KO mouse embryonic fibroblasts (MEF) were seeded onto three 150-mm culture dishes at a density of $1.5 \times 10^6$ cells and $3.75 \times 10^6$ cells per dish, respectively. Two days after seeding, the MEF were washed with 10 mL of PBS and incubated in 100 µg/mL cycloheximide diluted in PBS at 37 °C for 20 min. After incubation, the cells were scraped, transferred to a conical tube, and centrifuged at $750 \times g$ for 3 min. The supernatant was discarded and the pellets resuspended in 1 mL of lysis buffer (50 mM MOPS, 15 mM MgCl$_2$, 150 mM NaCl, 100 µg/mL cycloheximide, 0.5% Triton X-100, 1 mg/mL heparin, 0.2 U/µL RNase inhibitor, 2 mM phenylmethylsulfonyl fluoride (PMSF), and 1 mM benzamidine) and centrifuged. For fractionation, a sucrose gradient (10 mL of 10–50%) was established and the soluble fraction of the lysed sample was loaded onto the sucrose gradient and centrifuged at $160,000 \times g$ at 4 °C for 2 h (SW-41 Ti rotor, Beckman Coulter, Brea, CA, USA). After centrifugation, the gradients were fractionated using an ISCO tube piercer (Brandel, Gaithersburg, MD, USA) and collected using a fraction collector (Bio-Rad, Hercules, CA, USA). cDNA was synthesized from total RNA isolated from each fraction using M-MLV reverse transcriptase and the mRNA of target genes was amplified by PCR. Amplicons were visualised and assessed by agarose gel electrophoresis.

**Luciferase reporter gene assay**. The *Rhebl1* TBE-luciferase reporter construct was transfected into HEK293T cells with *Taz* or *Tead4*-expressing plasmids. A *Renilla* luciferase-expressing plasmid was also transfected in all experiments. For transfection, the Xtremegene 9 (Sigma-Aldrich, St. Louis, MO, USA) transfection reagent was used. After 24 h, the cells were harvested and the luciferase activity was measured using a Dual-Luciferase Reporter Assay (Promega, Madison, WI, USA). The firefly luciferase activity of *Rhebl1* TBE-luciferase was normalized to that of *Renilla* luciferase. The values of the experiments are presented as relative fold induction compared to the control condition.

**MitoTracker staining**. Cells were seeded on poly-L-lysine-coated coverslips and maintained in growth medium for 24 h. The cells were stained with 100 nM MitoTracker Red CMXROS (Invitrogen, Carlsbad, CA, USA) for 30 min. The cells were then washed twice with PBS, fixed in 3.7% formaldehyde for 15 min, permeabilised with ice-cold acetone for 5 min, and washed again with PBS. Cell nuclei were then counterstained with DAPI and mounted. The fluorescent signal was observed using a confocal microscope (LSM 510META, Carl Zeiss, Oberkochen, Germany). Zeiss LSM 5 software (v3.2) was used to acquire images and Zeiss LSM Image Examiner (v4,0,0,241) was used for fluorescence image processing.

**ATP bioluminescence assay**. C2C12 cells and MEF were seeded on culture plates. After 24 h of incubation, the cells were rinsed with 1× PBS and trypsinised. The cell number was counted and $1 \times 10^6$ cells were collected. All subsequent steps were performed using an ATP Bioluminescence HS II Kit (Roche), according to the manufacturer's protocol. Briefly, pellets were resuspended in 70 μL of dilution buffer. The same volume of cell lysis reagent was added to each sample followed by incubation at room temperature for 5 min. Then, 100 μL of each sample was transferred to a microfuge tube and the same volume of luciferase reagent was added. Luciferase activity was measured using a luminometer (GLOMAX, Promega, Madison, WI, USA).

**Oxygen consumption assay for cultured cells**. The real-time measurements of oxygen tension and cellular pH were obtained using an XF24 flux analyser (Seahorse Bioscience, North Billerica, MA, USA). The OCR was calculated from these measurements. One day before the assay, cells were seeded on an XF24 cell culture plate and the sensor cartridge was hydrated. All subsequent steps followed the manufacturer's protocol. During the experiment, cells were metabolically challenged using 1 μg/μL oligomycin, 0.5 μM carbonyl cyanide 4-(trifluoromethoxy) phenylhydrazone (FCCP), and 0.1 μM rotenone for bioenergetic profile assessment, namely, ATP turnover, natural proton leak of the mitochondrial inner membrane, and maximal respiratory capacity. After analysis, cells were counted for normalization. For the myotubes, genomic DNA was isolated from cells in each well and quantified for normalization.

**Isolation of muscle mitochondria**. For isolation of mitochondria from mouse skeletal muscle, Mitochondria Isolation Kit for Tissue (Thermo Fisher Scientific, 89801) was used. Experimental steps followed the manufacturer's protocol. In brief, isolated mouse gastrocnemius muscle was washed with PBS and cut into small pieces. After incubation in trypsin for 3 min, BSA/Reagent A was added and tissues were homogenized in a pre-chilled dounce homogenizer. Then, Reagent C was added and mixed by inverting. After centrifugation at $700 \times g$ for 10 min, the supernatant was transferred to a new tube. Samples were centrifuged again at $3,000 \times g$ for 15 min and the supernatant was discarded. Pellets were washed with Wash buffer and centrifuged at $12,000 \times g$ for 5 min. All steps were performed at 4 °C. Pellets were lysed with RIPA buffer for protein quantification or diluted in mitochondrial assay solution for oxygen consumption assays.

**Oxygen consumption assay for isolated mitochondria**. Isolated mitochondria were diluted in mitochondrial assay solution (70 mM sucrose, 220 mM mannitol, 5 mM $KH_2PO_4$, 5 mM $MgCl_2$, 2 mM HEPES, 1 mM EGTA, 0.2% FA-free BSA, pH 7.2) and seeded into XF24 cell culture plates. After centrifugation at $3000 \times g$, 4 °C for 20 min, 5.5 mM succinate and 2.2 μM rotenone in mitochondrial assay solution were added to each well. Plates were incubated at 37 °C (no $CO_2$) for 10 min. All subsequent steps followed the manufacturer's protocol. Mitochondria were metabolically challenged with 1 mM adenosine diphosphate (ADP), 4 μM oligomycin, 4 μM FCCP, and 4 μM antimycin A. Data were normalized to mitochondrial protein mass.

**ROS detection**. MEF were seeded on 96-well plates at $1 \times 10^4$ cells/$cm^2$. After 24 h, the cells were treated with 1 μM 5-(and-6)-chloromethyl-2′,7′-dichlorodihydro-fluorescein diacetate (CM-$H_2$DCFDA, an oxidative stress indicator; Invitrogen, Carlsbad, CA, USA) in PBS at 37 °C for 30 min. The probe was then removed and growth medium was added to cells which were then incubated for 30 min. Fluorescence was detected using a microplate reader (peak excitation = 495 nm, emission = 527 nm).

**Electron microscopy**. Tissue blocks ($1 \times 1 \times 1$ mm) were fixed in 4% formaldehyde and 1% glutaraldehyde in 0.1 M phosphate buffer (pH 7.4) overnight. The blocks were then immersed in 8% sucrose in 0.1 M phosphate buffer for 15 min. After three 15-min incubations in 8% sucrose, post-fixation was performed in 1% osmium tetroxide in 0.1 M phosphate buffer for 1 h. The tissue blocks were then dehydrated through an ethanol series and embedded with EMBed 812 (EMS #14120, Electron Microscopy Sciences, Hatfield, PA, USA) at 60 °C for 48 h. The embedded blocks were sectioned and collected on grids. The grids were stained with uranyl acetate and lead citrate and observed under a transmission electron microscope.

**Immunofluorescence**. For the paraffin-based method, mouse tissues were fixed in 4% paraformaldehyde and incubated at 4 °C for 48 h with rotation. The tissues were then dehydrated using an ethanol series, incubated in xylene, and embedded in paraffin at 60 °C overnight to make a paraffin tissue block. The paraffin block was then sectioned into 5-mm slices using a microtome (Leica, Wetzlar, Germany) and the sections were attached to glass slides. The tissue sections were rehydrated, followed by an antigen retrieval step. The sectioned tissues were then blocked using normal goat serum for 30 min and incubated with primary antibodies diluted in PBST (8 mM $Na_2HPO_4$, 150 mM NaCl, 2 mM $KH_2PO_4$, 3 mM KCl, 0.05% Tween 20, pH 7.4) at 4 °C overnight. After three washes with PBST, fluorophore-conjugated secondary antibodies diluted in PBST were added to the specimens

which were then incubated at room temperature for 2 h. The specimens were then washed again with PBST, mounted with DAPI-containing mounting medium (Vector Laboratories, Burlingame, CA, USA), and observed under a fluorescence microscope. Zeiss LSM 5 software (v3.2) was used to acquire images and Zeiss LSM Image Examiner (v4,0,0,241) was used for fluorescence image processing.

**Measurement of mouse metabolic parameters**. The Oxylet system from Panlab (Barcelona, Spain) was used for these measurements. The calorimetric chamber was assembled and the food/drink and gas ($O_2/CO_2$) analysers were calibrated. Mice were acclimated in a calorimetric home cage for 3 days and data were collected for 1 day. During data collection, $VO_2$ and $VCO_2$ were measured every 15 min. Energy expenditure was calculated from the collected data using META-BOLISM software (Panlab, Barcelona, Spain). To analyse animal activity, rearing and movement were detected using an infrared-based sensor frame and weight transducer, respectively. METABOLISM software (Panlab, v3.0) was used to collect and analyse data.

**Rotarod test**. Mice were acclimated to room conditions for 1 day and then loaded onto a rotarod treadmill (ENV-574M, Med Associates, Fairfax, VT, USA) for testing. The speed of rotation was increased from 4 rpm to 40 rpm over 3 min and then 40 rpm was maintained until the end of the experiment. Experiments were performed once a day, for 3 days, in the dark.

**Treadmill endurance test**. Mice were familiarized with the training program (16 m/min treadmill velocity, 10 min training time, 5° slope angle) for 2 days. After familiarization, an endurance test was commenced using the trained mice. The endurance test was started at a velocity of 10 m/min, which was gradually increased with 2 m/min increments every 10 min, up to a maximum of 20 m/min. Fatigue was defined as a state in which the mice did not reengage the treadmill and remained on the shock grid for 5 s. Running distance was calculated from the velocity values and the running time of each mouse. Exhausted mice were immediately removed from the treadmill and allowed to rest. The test was performed during the daytime.

**Quantification of fluorescence signal and transmission electron microcopy images**. Fluorescence signal intensity and mitochondria number/area of transmission electron microcopy images were measured using ImageJ (1.43 u) software (NIH).

**Statistical analysis**. For the in vivo experiments, all data are given as the mean ± standard error of the indicated sample number. For the in vitro experiments, data are given as the mean ± standard deviation of at least three independent experiments. The results were analysed for statistical significance by unpaired Student's $t$-test, one-way or two-way analysis of variance (ANOVA) with Tukey's and Sidak's multiple comparison test. The level of significance is indicated in figure legend with the number of asterisks and $p$-values. Microsoft Excel (v16.0) and GraphPad Prism 6 (v6.01) were used for statistical analysis.

**Reporting summary**. Further information on research design is available in the Nature Research Reporting Summary linked to this article.

## Data availability

All data are provided in the paper and/or Source data files and Supplementary Information. The following database was used to search TAZ-binding elements in *Rhebl1* enhancer in this study: ArrayExpress database (https://www.ebi.ac.uk/arrayexpress/experiments/E-MTAB-6764/). Source data are provided with this paper.

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

## Acknowledgements

This work was supported by the Basic Science Research Program of NRF funded by the Ministry of Education (2019R1A2C2008430, 2021R1A2C1007461, 2020R1A2C2004679, and 2018R1A5A2025286), Republic of Korea. This work was also supported by a grant offered by Korea University.

## Author contributions

J.H. Hwang performed the experimental work, analysed the data, and drafted the manuscript. K.M.K., H.T.O., G.D.Y., M.G.J., H.L., J.P., and K.J. prepared the experimental reagents, set up the experimental system, and contributed to the discussions of the experimental data. Y.K.K. and Y.G.K. contributed to the discussions of the experimental data. E.S.H. and J.H. Hong led the project, interpreted the data, and drafted the manuscript.

## Competing interests

The authors declare no competing interests.
