## [Peer Review File · Nature Communications]

REVIEWER COMMENTS

Reviewer #1 (Remarks to the Author):

This is an interesting manuscript revealing a role for Taz in regulating Tfam translation and exercise-induced mitochondrial biogenesis.

Major comments:

1. Many statements about how "Taz stimulates...", or "Taz induces..." but this seems to imply that the addition of Taz would be sufficient to induce the response referred to. The authors have not used any over-expression, other than for rescue, to show that Taz is sufficient to cause mitochondrial biogenesis. I think the wording in these statements should be changed because they are misleading.
2. Many results report the "transcriptional effect", when in fact transcript levels are measured only. The mRNA could reflect changes in mRNA decay as well as transcription. Please change the wording to "transcript level, or equivalent".
3. In figure 1: the loss of mitochondrial content in Tax KO animals could simply be a function of a lower inherent activity level of the animals. If they are less active, they will have less mitochondria. Please add this as a possibility or comment in the discussion.
4. There is discussion of Yap and Hippo signaling, but it is not clear how this fits into the story presented. Blots of Yap are shown, yet none of this is discussed in the context of the summary diagram in Ext Fig. 5. This could be put into context more effectively.
5. In the absence of Taz, exercise is not capable of inducing mtDNA transcripts, presumably as a result of a lack of Tfam. This would produce mitochondria with altered stoichiometric ratios of ETC subunits, and it is not clear that the mitochondria would be functioning correctly. In my view, verification of accurate "mitochondrial biogenesis" requires in this case an assessment of mitochondrial function following training (ROS, respiration);
6. The absence of Taz also affects the nuclear genome, as shown in Fig. 1 by the reduced levels of SDH, COXIV, cyto c. Clearly it does not operate solely via Tfam activity on mtDNA. This should be discussed.
7. The abstract contains information on IL-6 effects, yet this is positioned in the extended data. Seems like it should be more up-front, if it is to be a main conclusion of the paper;

8. The hypothesis that Taz operates in the post-exercise window may be overstated, as the increases in PGC-1 are also exaggerated post-exercise. Indeed, recovery is very important for exercise-induced adaptations, either involving Taz or other transcriptional regulators.

Reviewer #2 (Remarks to the Author):

They show that TAZ KO in skeletal muscle impairs mitochondrial biogenesis, respiratory metabolism, and response to exercise. The exciting implication here is that TAZ may be a "new" PGC1a for mitochondrial biogenesis. The topic is interesting and novel.

However, a limitation of this work is that it deals with a developmental phenotype. I would have preferred much more a conditional KO of TAZ in adult muscle. Lack of temporal control of when and where the defect initiates is a severe negative because what they observe may be the consequence of a series of events. What they show is a generalized reduction of metabolic fitness, that fits with a generic notion that YAP/TAZ are relevant for metabolism. Of course here they point to novel aspects, that may not even be so specific for muscle, but general, given the similarity of effects shown in KO mef and even in the myoblasts line C2C12.

At the same time, their work in vitro on TAZ KD and MEF KO is clean and convincing, alleviating the above.

What is instead entirely missing is the validation of the TAZ-Rheb-mTORC1 axis in vivo.

They should show phenocopy and/or TAZ KO rescue in more in vivo settings. Using drug treatments, or electroporations in muscles, or (obviously less feasible) combined transgenic mice. There is in other words a gap too deep between their in vivo conditions and mechanistic claims in vitro. They should go back and reconfirm functionally the relevance of the mechanism - at least some of it! - and not just through correlative lines of evidence as shown so far.

This would be plenty for a round story in a revised paper. Please take out the IL6 that would spread the story even thinner, and is really very poorly and loosely connected, primordial and, in case, may deserve a story on its own. Of course the paper raises a number of exciting questions that cannot fit into this publishable unit, but may open new avenues of research.

Minor. TAZ should be explicit in the title instead of its real name.

Responses to Reviewer 1's Comments

This is an interesting manuscript revealing a role for Taz in regulating Tfam translation and exercise-induced mitochondrial biogenesis.

1. Many statements about how "Taz stimulates..." or "Taz induces..." but this seems to imply that the addition of Taz would be sufficient to induce the response referred to. The authors have not used any overexpression, other than for rescue, to show that Taz is sufficient to cause mitochondrial biogenesis. I think the wording in these statements should be changed because they are misleading.

Response: To avoid the possibility of being misleading, we have now edited the statements. Some instances of "TAZ" have been replaced with "endogenous TAZ" to clearly address that endogenous TAZ protein stimulates or induces target genes. The edited sites are highlighted in the revised manuscript.

2. Many results report the "transcriptional effect", when in fact transcript levels are measured only. The mRNA could reflect changes in mRNA decay as well as transcription. Please change the wording to "transcript level, or equivalent."

Response: As the reviewer recommended, we have changed the relevant section. The edited sites are highlighted in the revised manuscript.

3. In figure 1: the loss of mitochondrial content in Taz KO animals could simply be a function of a lower inherent activity level of the animals. If they are less active, they will have less mitochondria. Please add this as a possibility or comment in the discussion.

Response: Thank you for the comment. To address whether the diminished activity of TAZ mKO mice affects mitochondrial content, we generated TAZ knockdown mice by the introduction of TAZ shRNA carrying adeno-associated virus into adult muscle. Because the TAZ shRNA-carrying virus was injected into a leg muscle and control virus was injected into the contralateral leg muscle in the same mouse, we were able to study mitochondrial biogenic effects without considering a lower inherent activity level of mice. In Supplementary Fig. 2 of the revised manuscript, we observed that TAZ knockdown muscle exhibited decreased

mitochondrial mass compared to control muscle. Therefore, these results suggested that the difference in mitochondrial mass between WT and TAZ mKO mice was not caused by lower inherent activity.

Supplementary Figure 2

4. There is discussion of Yap and Hippo signaling, but it is not clear how this fits into the story presented. Blots of Yap are shown, yet none of this is discussed in the context of the summary diagram in Ext Fig. 5. This could be put into context more effectively.

Response: Thank you for these comments. To study the effect of YAP in mitochondrial biogenesis, YAP knockdown was assessed in C2C12 myotubes. From the bottom Extended Data Fig.1, YAP knockdown caused no decrease in the levels of Rheb11, Rheb, phosphorylated 4E-BP1, or phosphorylated S6K. In addition, mitochondrial marker gene expression and mitochondrial DNAs were not altered after YAP knockdown. Thus, our results suggested that YAP knockdown had no effect on mitochondrial biogenesis *in vitro*. However,

because the *in vivo* effect of YAP was not investigated in this study, we could not clearly address the role of YAP in the summary diagram.

Extended Data Fig. 1. YAP depletion showed no differences in Rheb/Rheb1 level and mitochondrial mass in C2C12 myotube. **a**) Protein was isolated from the control (Con) and YAP knockdown (KD) C2C12 myotubes, and assessed via immunoblotting. Vinculin was used as the loading control. **b**) RNA was isolated from the Con and YAP KD C2C12 myotubes and the transcripts level of Rheb and Rheb1 were analyzed by qRT-PCR. **c**) RNA from the Con and YAP KD C2C12 myotubes was assessed by qRT-PCR to determine the transcripts level of mitochondria-encoded genes. **d**) Genomic DNA was isolated from the Con and YAP KD C2C12 myotubes and relative mitochondrial DNA copy number was determined via quantitative PCR using primers for mitochondrial-encoded *Cox2* and nuclear-encoded *β -globin*. For panel **b** to **d**, the experiments were performed in triplicates.

5. In the absence of Taz, exercise is not capable of inducing mtDNA transcripts, presumably as a result of a lack of Tfam. This would produce mitochondria with altered stoichiometric ratios of ETC subunits, and it is not clear that the mitochondria would be functioning

correctly. In my view, verification of accurate "mitochondrial biogenesis" requires in this case an assessment of mitochondrial function following training (ROS, respiration).

Response: As the reviewer suggested, we isolated mitochondria from exercise-trained WT and mKO mice and analysed oxygen consumption rate via XF analyser. The results have been added into Supplementary Fig. 5. The results showed that mitochondria isolated from muscle of TAZ knockout function, but have decreased respiratory activity in both untrained and trained conditions.

Supplementary Figure 5

6. The absence of Taz also affects the nuclear genome, as shown in Fig. 1 by the reduced levels of SDH, COXIV, Cyto c. Clearly it does not operate solely via Tfam activity on mtDNA. This should be discussed.

Response: Thank you for the constructive comment. The reviewer's point is correct. TAZ depletion also further decreased the reduced levels of SDH, COXIV, and Cyto C, which are not regulated by Tfam. In addition to the induction of *Tfam* mRNA translation by TAZ, we also confirmed translational inhibition of another mitochondrial gene transcript, *Atp5d* (Fig. 2k). As reported by the Soneneberg group (Larsson O. et al, 2012, PNAS; Morita M. et al, 2013, Cell Metabolism), mTORC1 inhibition suppresses translation of a variety of mitochondrial gene transcripts including *Tfam* and *Atp5d*. Therefore, other nuclear-encoded mitochondrial gene transcripts such as *SDH*, *COXIV*, and *CytoC* may be regulated by TAZ at the level of translation through mTORC1 signalling. We now canvas this possibility in the Discussion section.

7. The abstract contains information on IL-6 effects, yet this is positioned in the extended data. Seems like it should be more up-front, if it is to be a main conclusion of the paper;

Response: Because the effects of IL-6 are not a main consideration of this manuscript, as another reviewer recommended, the results were eliminated in the revised manuscript.

8. The hypothesis that Taz operates in the post-exercise window may be overstated, as the increases in PGC-1 are also exaggerated post-exercise. Indeed, recovery is very important for exercise-induced adaptations, either involving Taz or other transcriptional regulators.

Response: I agree with the reviewer, TAZ is one of the proteins which operated in post-exercise. We tried to avoid overstating the function of TAZ in Results and Discussion. We edited the manuscript and edited sites are highlighted in the revised manuscript.

Responses to Reviewer 2's Comments:

1. They show that TAZ KO in skeletal muscle impairs mitochondrial biogenesis, respiratory metabolism, and response to exercise. The exciting implication here is that TAZ may be a "new" PCG1a for mitochondrial biogenesis. The topic is interesting and novel.

However, a limitation of this work is that it deals with a developmental phenotype. I would have preferred much more a conditional KO of TAZ in adult muscle. Lack of temporal control of when and where the defect initiates is a severe negative because what they observe may be the consequence of a n series of events. What they show is a generalized reduction of metabolic fitness, that fits with a generic notion that YAP/TAZ are relevant for metabolism. Of course here they point to novel aspects, that may not even be so specific for muscle, but general, given the similarity of effects shown in KO mef and even in the myoblasts line C2C12. At the same time, their work in vitro on TAZ KD and MEF KO is clean and convincing, alleviating the above.

Response; Thank you for these constructive comments. To confirm the effect of muscle-specific depletion of TAZ in adult mice, we used adeno-associated virus (AAV) to carry shRNA for TAZ knockdown. Through viral transduction into gastrocnemius muscle, we observed significant downregulation of mitochondrial mass as evidenced by decreased

mitochondrial DNA copy number, mitochondrial marker gene transcripts and protein levels, and fluorescent signal of mitochondrial protein (Supplementary Fig. 2). TAZ knockdown was confirmed by immunoblot assays, as described in Supplementary Fig. 2b.

Supplementary Figure 2

2. What is instead entirely missing is the validation of the TAZ-Rheb-mTORC1 axis in vivo. They should show phenocopy and/or TAZ KO rescue in more in vivo settings. Using drug treatments, or electroporations in muscles, or (obviously less feasible) combined transgenic mice. There is in other words a gap too deep between their in vivo conditions and mechanistic claims in vitro. They should go back and reconfirm functionally the relevance of the mechanism - at least some of it! - and not just through correlative lines of evidence as shown so far.

Response; Thank you for your constructive comments. We rescued Rheb1 expression in gastrocnemius muscle of adult mouse by AAV and observed restoration of mitochondrial mass (Fig. 4). Rescued Rheb1-mTOR signalling was verified by immunoblot assays, as

depicted in Fig. 4b. In addition, we stimulated mTOR signalling in TAZ mKO mice by injecting a mTOR stimulator, MHY1485. We observed recovery of mitochondrial gene transcription and mitochondrial DNA (Supplementary Fig. 4). Thus, these results suggest that TAZ is an important regulator in mTOR-mediated mitochondrial biogenesis.

Figure 4

Supplementary Figure 4

3. This would be plenty for a round story in a revised paper. Please take out the IL6 that would spread the story even thinner, and is really very poorly and loosely connected, primordial and, in case, may deserve a story on its own. Of course the paper raises a number of exciting questions that cannot fit into this publishable unit, but may open new avenues of research.

Response; As suggested, a part of the IL6 study was removed. Thank you for your constructive advice.

4. Minor. TAZ should be explicit in the title instead of its real name.

Response; As reviewer recommended, the title has been edited.

REVIEWER COMMENTS

Reviewer #1 (Remarks to the Author):

I have only one other comment: Is it fair to keep the title as it is, when it is evident that Taz activates mitochondrial biogenesis via the nuclear genome as well? Please consider. I do not need to see the decision.

Otherwise, all other comments have been addressed.

Reviewer #2 (Remarks to the Author):

I would have preferred some better genetics. Inducible Drivers are available after all. Anyway, the use of shRNA has improved the MS. The experiments however need better controls: add back with wild-type TAZ (to rescue and thus showing specificity of the only used shRNA) and ideally add back of a Taz point mutant (transcriptionally defective) unable to bind tead, that should not rescue.

For the rest I am ok with the MS

Responses to Reviewer 1's Comments

I have only one other comment: Is it fair to keep the title as it is, when it is evident that Taz activates mitochondrial biogenesis via the nuclear genome as well? Please consider. I do not need to see the decision.

Otherwise, all other comments have been addressed.

Response: Thank you for the comments. Our study reveals that mitochondrial transcription factor A is a noble target of TAZ, which stimulates mitochondrial gene transcription. Though the reviewer point is reasonable, we would like to stay the original title to emphasize our observation.

Responses to Reviewer 2's Comments:

I would have preferred some better genetics. Inducible Drivers are available after all. Anyway, the use of shRNA has improved the MS. The experiments however need better controls: add back with wild-type TAZ (to rescue and thus showing specificity of the only used shRNA) and ideally add back of a Taz point mutant (transcriptionally defective) unable to bind tead, that should not rescue.

For the rest I am ok with the MS

Response: Thank you for your constructive comments. As the reviewer recommended, we rescued wild-type TAZ or Tead binding mutant of TAZ in TAZ knockdown mice (Supplementary Fig. 2d). Then, we observed that wild-type TAZ recovers mitochondrial protein level and DNA copy number, but the recovery was not significant with the Tead binding mutant of TAZ (Supplementary Fig. 2e and 2f).

Supplementary Figure 2